

# Causal graph extraction from news: a comparative study of time-series causality learning techniques

Mariano Maisonnave[1,2], Fernando Delbianco[3,4], Fernando Tohme[3,4], Evangelos Milios[2] and Ana G. Maguitman[1,5]

[1] Departamento de Ciencias e Ingeniería de la Computación, Universidad Nacional del Sur, Bahía Blanca, Buenos Aires, Argentina
[2] Faculty of Computer Science, Dalhousie University, Halifax, Canada
[3] Instituto de Matemática de Bahía Blanca, Bahía Blanca, Buenos Aires, Argentina
[4] Departamento de Economía, Universidad Nacional del Sur, Bahía Blanca, Buenos Aires, Argentina
[5] Instituto de Ciencias e Ingeniería de la Computación (UNS-CONICET), Bahía Blanca, Buenos Aires, Argentina

Corresponding authors
Mariano Maisonnave,
mariano.maisonnave@cs.uns.edu.ar
Ana G. Maguitman,
agm@cs.uns.edu.ar

## ABSTRACT

Causal graph extraction from news has the potential to aid in the understanding of complex scenarios. In particular, it can help explain and predict events, as well as conjecture about possible cause-effect connections. However, limited work has addressed the problem of large-scale extraction of causal graphs from news articles. This article presents a novel framework for extracting causal graphs from digital text media. The framework relies on topic-relevant variables representing terms and ongoing events that are selected from a domain under analysis by applying specially developed information retrieval and natural language processing methods. Events are represented as event-phrase embeddings, which make it possible to group similar events into semantically cohesive clusters. A time series of the selected variables is given as input to a causal structure learning techniques to learn a causal graph associated with the topic that is being examined. The complete framework is applied to the New York Times dataset, which covers news for a period of 246 months (roughly 20 years), and is illustrated through a case study. An initial evaluation based on synthetic data is carried out to gain insight into the most effective time-series causality learning techniques. This evaluation comprises a systematic analysis of nine state-of-the-art causal structure learning techniques and two novel ensemble methods derived from the most effective techniques. Subsequently, the complete framework based on the most promising causal structure learning technique is evaluated with domain experts in a real-world scenario through the use of the presented case study. The proposed analysis offers valuable insights into the problems of identifying topic-relevant variables from large volumes of news and learning causal graphs from time series.

## INTRODUCTION

Causal modeling aims to determine the cause–effect relations among a set of variables. A variable is the basic building block of causal models and represents a property or descriptor that can take multiple values (*Glymour, Pearl & Jewell, 2016*). The extraction of variables and their causal relations from news has the potential to aid in the understanding of complex scenarios. In particular, it can help explain and predict events, as well as conjecture about possible causality associations. Although the problem of causal modeling has attracted increasing attention in the Computer Science discipline (*Pearl, 2009*; *Bareinboim & Pearl, 2015*; *Peters, Janzing & Schölkopf, 2017*; *Meinshausen et al., 2020*), limited work has been devoted to the problem of large-scale extraction of causal graphs from news articles. Causality can provide tools to better understand machine learning models and their applicability. However, black-box predictive models have typically dominated machine learning-based decision making, with a lack of understanding of cause–effect connections (*Rudin & Radin, 2019*). On the other hand, causality has been central to econometrics, where most methods rely either on the analysis of structural models (*Heckman & Vytlacil, 2007*) or on the application of Granger's idea of causation based on determining whether past values of a time series provide unique information to forecast future values of another (*Granger, 1969*).

To discover causal relations, interventions and manipulations are typically necessary as part of a randomized experiment. However, undertaking such experiments is usually impractical or even impossible. As a consequence, to address these limitations, many methods for causal discovery usually rely on observational data only and a set of (strong) assumptions. The relatively recent availability of large volumes of data compensates to a certain degree for the infeasibility of experimentation, offering an opportunity to collect and exploit observational data for causal modeling and analysis (*Varian, 2014*).

Causality extraction from text has been previously explored mostly as a relation extraction problem, which can be addressed as a specific information extraction task. Existing approaches typically rely on the use of lexico-syntactic patterns (*Joshi et al., 2010*), supervised learning (*Khetan et al., 2022*), and bootstrapping (*Heindorf et al., 2020*). These approaches apply local analysis methods to extract explicit causal relations from text by adopting an intra- or inter-sentence scheme. However, these methods are unable to detect implicit causal relations that can be inferred from the analysis of time series data built from sentences coming from several documents. Also, due to the limited availability of ground truth for causal discovery, few studies have been carried out in the context of a real-world application. Finally, another limitation of previous approaches to causal extraction from text is the absence of a clear semantics associated with the variables that represent the cause–effect relations. In other words, variables are usually terms identified in text, with no distinction between general terms and variables built from event mentions.

The work presented in this article attempts to overcome these limitations. It proposes a methodology for causal graph extraction from news and presents comparative studies that allow to address the following research questions:

- **RQ1.** What state-of-the-art methods for time-series causality learning are effective in generalized synthetic data?
- **RQ2.** Which of the most promising methods for time-series causality learning identified through **RQ1** are also effective in real-world data extracted from news?
- **RQ3.** What type of variables extracted from a large corpus of news is effective for building interpretable causal graphs on a topic under analysis?

The proposed approach combines methods coming from information retrieval, natural language processing, machine learning, and Econometrics into a framework that extracts variables from large volumes of text to build highly interpretable causal models. The extracted variables represent terms (unigrams, bigrams and trigrams) and ongoing event clusters. The terms are selected from topic-relevant sentences using a supervised term-weighting scheme proposed and evaluated in our previous work (*Maisonnave et al., 2021a*; *Maisonnave et al., 2020*). In the meantime, the ongoing event clusters are computed by clustering event phrase embeddings, where the task of detecting ongoing events is defined and evaluated by the authors in *Maisonnave et al. (2021b)*. A time series of the selected variables is used to learn a causal graph associated with the topic that is being examined. The framework is applied to a case study using real-world data extracted from a 246-month period (roughly 20 years) of news from the New York Times (NYT) corpus (*Sandhaus, 2008*). To answer research question **RQ1** an evaluation based on synthetic data from *TETRAD* (*Scheines et al., 1998*) and *CauseMe* (*Runge et al., 2019a*) is carried out to gain insight into the selection of time-series causality learning techniques. This evaluation comprises a systematic analysis of nine state-of-the-art causal structure learning techniques and two ensemble methods derived from the most effective techniques. To address **RQ2** the proposed framework applies the most promising methods identified through **RQ1** to extract causal relations from news on a topic under analysis. Then, a comparative study of the candidate causal learning methods is conducted based on assessments provided by domain experts. Finally, to answer **RQ3** the two types of variables extracted by the framework are analyzed, namely general terms (unigrams, bigrams, and trigrams) and ongoing event clusters. Then, an evaluation of causal relations containing each type of variables is performed based on assessments derived from experts. This allows investigating whether there tends to be more agreement among experts when the variables representing potential causes and effects are of a specific type. It also allows determining if the evaluated causal extraction methods are more effective if the analysis is restricted to a certain type of variables.

Overall, the contributions of this work can be summarized as follows:

- A framework that combines term selection and event detection to build a time series that is used to learn causal models from large volumes of text. The framework introduces a novel method for building event-phrase embeddings, which groups events extracted from news into semantically cohesive event clusters.
- An extensive evaluation on synthetic data of nine state-of-the-art causal structure learning techniques and two novel ensemble techniques derived from the most effective ones.

- The application of the presented framework to a case study that allows to illustrate the proposed causal graph extraction methodology and to further evaluate the analyzed causality learning techniques in a real-world scenario. Also, as a byproduct of the evaluation, we offer a dataset consisting of domain expert causality assessments on pairs of variables extracted from real-world data.

The data and full code of the methods used by the framework and to carry out the experiments are made available to allow reproducibility.

## RELATED WORK

In Computer Science, approaches to causality have been mostly centered around probabilistic graphical models (*Koller & Friedman, 2009*), which are graphical representations of data and their dependency relationships. Bayesian networks (*Pearl, 2009*) are a kind of probabilistic graphical model used for causal inference by capturing both conditionally dependent and conditionally independent relations between random variables by means of a directed acyclic graph (DAG). Meanwhile, the study of the concept of causality is a central and long-standing issue in the field of Econometrics, where it has been addressed mainly by methods derived either from the analysis of structural models (*Heckman & Vytlacil, 2007*) or the application of the *Granger Causality test* (*Granger, 1969*). Both approaches are based on two principles: (1) a cause precedes the effect, and (2) the cause produces unique changes in the effect, so past values of the cause help predict future values of the effect. In the case of causal structure models, different techniques have been developed, which are typically classified into three main categories, namely (1) independence-based causal structure learning (*Spirtes & Glymour, 1991*; *Runge et al., 2019b*), (2) restricted structural causal models (*Shimizu et al., 2006*; *Shimizu et al., 2011*), and (3) autoregressive models (*Granger, 1969*; *Schreiber, 2000*; *Sims, 1980*; *Nicholson, Matteson & Bien, 2017*; *Chiquet et al., 2008*). Autoregressive models are defined exclusively for time series, where lagged variables(*i.e.,* dependent variables that are lagged in time) play a key role. It is worth mentioning that if we combine time-lagged and non-time-lagged variables, the autoregressive approach can be seen also as an independence-based approach with respect to the non-time-lagged variables.

Several previous works have addressed the problem of causal structure learning from text. *Silverstein et al. (2000)* propose a series of algorithms that combine different heuristics to identify causal relationships from heterogeneous datasets. In particular, the algorithms were run on large volumes of text from news that cover different topics and have demonstrated to have the capability of efficiently returning a number of causal relationships and not-directly-causal relationships. Another approach for extracting causal relations from text was presented by *Girju & Moldovan (2002)*, where a semi-automatic method is proposed to identify lexico-syntactic patterns referring to causation. A system for acquiring causal knowledge from text was proposed by *Sanchez-Graillet & Poesio (2004)*. The system identifies sentences that specify causal relations and builds Bayesian networks by extracting causal patterns from the sentences. *Dehkharghani et al. (2014)* proposed a method for causal rule discovery that combines sentiment analysis and association rule

mining. Another work proposed in *Heindorf et al. (2020)* extracts *claimed* causal relations from the Web to induce the *CauseNet* causality graph, containing approximately 200,000 relations. The above works take linguistic or data mining approaches, where certain syntactic regularities that are manually crafted or automatically generated using machine learning techniques allow to detect pairs of terms potentially related by a causal relation. Since many of these contributions rely on detecting explicit patterns or causality cues in texts, their focus is on inferring the existence of causality links that are explicit in continuous spans of text. Since our approach is based on building time series on term and event frequencies throughout the whole dataset, we can detect *implicit* causal links between words that are not even present in the same text span (or article). A recent survey that reviews techniques for the extraction of explicit and implicit inter- and intra-sentential causality from natural language text is presented by *Yang, Han & Poon (2021)*.

The framework described in this article is closely related to the one presented by *Radinsky, Davidovich & Markovitch (2012)*, where semantic natural language modeling, machine learning, and data mining techniques are applied to 150 years of news articles to identify causal predictors of events. In that work, the authors apply semantic natural language modeling techniques to titles containing certain predefined causality patterns. This allows them to find causal links that are not explicit in texts using generalizations supported by a vast amount of world knowledge ontologies mined from Linked Data (*Bizer, Heath & Berners-Lee, 2011*). By using hand-crafted rules supported by these ontologies, the authors are able to extract causality pairs with high precision. However, this is gained at the expense of recall, which is only 10%. Although the authors find additional causality patterns through generalization, they still need to detect mentions to causality to trigger generalizations. In contrast to that contributions, we do not rely on ontologies, detection of semantic pattern, or explicit mentions of causality in texts. Also, different from our approach, the focus of *Radinsky, Davidovich & Markovitch (2012)* is not the identification and extraction of causality but the prediction of future events caused by a given event.

Another related work was presented by *Balashankar et al. (2019)*, where the authors describe a framework that allows to build a predictive causal graph by measuring how the occurrence of a word in the news influences the occurrence of other words in the future supported by the concept of Granger Causality. This work is closely related to ours. However, we identified several limitations in their contribution, which we address in our work. First, the authors only apply one technique of causality detection, *i.e.,* the Granger Causality test. Second, since these authors work with existing methods for detecting event triggers based on the ACE 2005 task description (*Walker et al., 2006*), their work does not incorporate the concept of ongoing events. Lastly, because their focus is on stock price prediction, they are not concerned with explaining or evaluating the resulting causal graphs.

Here, we extend the application of causal structure learning techniques to uncover relations among variables representing terms and events extracted from digital media, aiming to detect a network of causal links among these variables. Because our approach does not require explicit mentions to causality or the identification of semantic patterns and does not require ontologies either, it could be easily applied to different domains with little

or no modification. Even more, our framework could be generalized to other languages as long as a model for ongoing event detection in the required language is available. We believe our work is especially well-suited as a tool to understand complex scenarios or topics using only a set of texts describing them.

# AN OVERVIEW OF CAUSAL STRUCTURE LEARNING

Causal learning is the process of inferring a causal model from data (*Peters, Janzing & Schölkopf, 2017*) while causal structure learning is the process of learning the causal graph or certain aspects of it (*Heinze-Deml, Maathuis & Meinshausen, 2018*). In this work, we address the causal structure learning problem, where data are presented as a time series of variables that stand for terms or events. A variety of techniques have been proposed in the literature to address causal structure learning. This section outlines and evaluates nine state-of-the-art and two ensemble techniques for causal structure learning from time series of independent and identically distributed random variables. The goal of this analysis is to identify the most promising techniques with the purpose of incorporating them into the proposed causal learning framework.

The analyzed techniques for causal structure learning are classified into three main categories, namely (1) independence-based causal structure learning, (2) restricted structural causal models, and (3) autoregressive models. A general overview of the analyzed causal structure learning techniques is presented in Fig. 1.

Independence-based causal structure learning relies on two main assumptions: *the Markov property for directed graphs* and *faithfulness* (*Koller & Friedman, 2009*). These assumptions allow estimating the Markov equivalence classes of the DAG from the observational data. All DAGs in an equivalence class have the same skeleton (*i.e.,* the causal graph with undirected edges) and the same v-structures (*i.e.,* the same induced subgraphs of the form $X \rightarrow Y \leftarrow Z$). However, the direction of some edges may not be determined. Since this work analyzes non-contemporary causalities, the direction of time can be used to determine the direction of the edges that remain undirected. That is, since the cause has to happen before the effect, it is known that the arrows cannot go back in time. Using this criterion, a fully directed graph is obtained from the independence-based techniques used. For the present work, we consider *PC* (*Spirtes & Glymour, 1991*) and *PCMCI* (*Runge et al., 2019b*) as two representative techniques based on independence. The *TIGRAMITE* package (https://github.com/jakobrunge/tigramite) (*Runge et al., 2019b*) is used to evaluate both techniques and to analyze the conditional independencies in the observed data through a partial correlation test (*ParCor*). This test estimates the partial correlations by means of a linear regression computed with ordinary least squares and a non-zero Pearson linear correlation test in the residuals. Other non-linear conditional independence tests are not included because of their prohibitive computation time.

The techniques based on restricted structural causal models incorporate additional assumptions to obtain identifiability. For instance, in the case of non-Gaussian linear models (LiNGAM), it is possible to analyze the asymmetry between cause and effect to distinguish cause from effect. This is possible because the regression residuals are
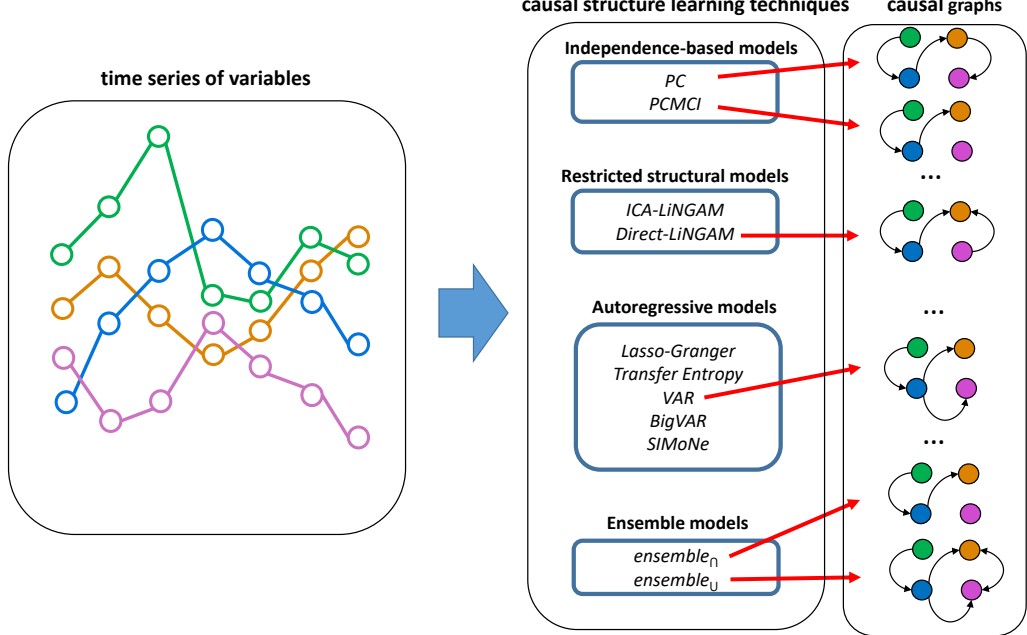

**Figure 1** **Overview of time series causal structure learning techniques analyzed in this work.** These techniques take a time series of variables and generate a causal graph. The analyzed techniques are divided into independence-based models, restricted structural models, autoregressive models, and ensemble models. The ensemble models combine *Direct-LiNGAM*, *PCMCI*, *VAR*, and *PC* (which proved to be the four most effective techniques according to evaluations carried out on synthetic data).

independent of the predictor only for the correct causal direction. This work analyzes two techniques based on restricted structural models, namely *ICA-LiNGAM* (*Shimizu et al., 2006*) and *Direct-LiNGAM* (*Shimizu et al., 2011*).

The techniques based on autoregressive models are defined exclusively for time series and are based on determining whether past values of a variable $X$ offer unique information (*i.e.,* not provided by other variables) to predict or explain future values of another variable $Y$. If this is the case, it is possible to hypothesize that $X \rightarrow Y$. This idea gives rise to the statistical concept of causality known as *Granger Causality* (*Granger, 1969*). This work analyzes five techniques for inferring causal structures in time series based on these principles: (1) *Lasso-Granger* (*Granger, 1969*), (2) *Transfer Entropy* (*Schreiber, 2000*), (3) *VAR* (*Sims, 1980*), (4) *BigVAR* (*Nicholson, Matteson & Bien, 2017*), and (5) *SIMoNe* (*Chiquet et al., 2008*).

Two ensemble techniques are also implemented by combining the four most effective state-of-the-art causal structure learning techniques (*Direct-LiNGAM*, *PCMCI*, *VAR*, and *PC*) based on the evaluations carried out on synthetic data (to be presented in the next section). The first ensemble technique, referred to as *ensemble*$_\cap$, adds a causal relation only when the four best techniques agree on including it. On the other hand, the second ensemble technique, called *ensemble*$_\cup$, adds a causal relation when any of the four techniques includes it. Finally, for the sake of comparison, a baseline model, referred to
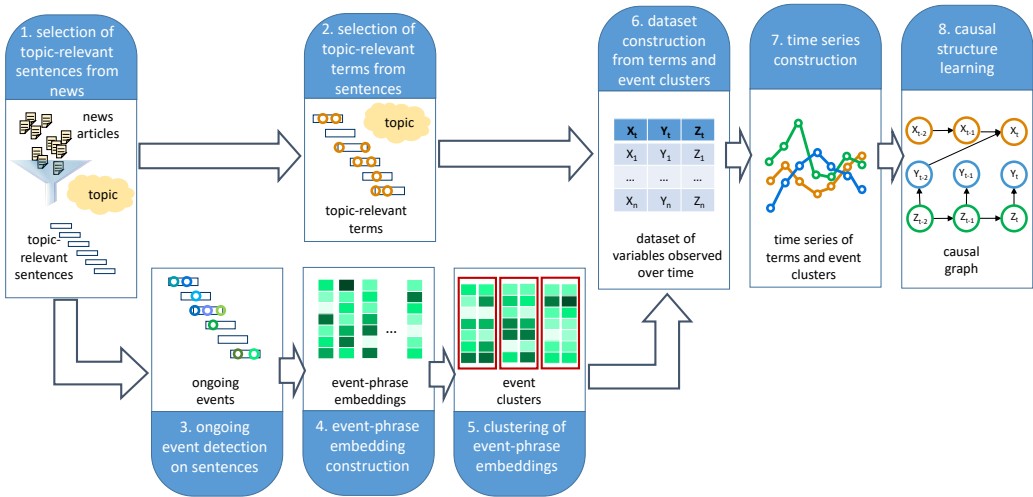

**Figure 2** **Framework for causal graph extraction from digital media.** The framework takes as input a topic description and a corpus of news articles. It then applies eight steps aimed at building a causal graph associated with the topic of interest.

as *Random* is also considered. The *Random* technique decides on a random basis with probability 0.5 whether to add or not each potential edge to the graph.

# A FRAMEWORK FOR CAUSAL LEARNING FROM NEWS

This section describes the proposed framework, which provides support to experts while trying to analyze a specific topic by semi-automatically identifying relevant variables associated with a given topic and suggesting potential causal relations among these variables to build a causal graph. A diagram of the framework for building a causal graph from digital text media is presented in Fig. 2. The framework completes the following steps:

- **Step 1. Filter topic-relevant sentences from a collection of news.** Given a topic description the framework selects those sentences that match the topic. A topic can be represented in a variety of ways. Examples of simple topic representations are n-grams or sets of n-grams. However, more complex schemes for representing topics can be naturally adopted by the framework, including machine-centered representations, such as vector space models or multimodal mixture models, and human-centered representations, such as concept maps.

- **Step 2. Select terms from the topic-relevant sentences.** The selection of relevant terms (unigrams, bigrams, and trigrams) from the given sentences relies on FDD$_\beta$, a supervised term-weighting scheme proposed and evaluated by the *Maisonnave et al. (2021a)* and *Maisonnave et al. (2020)*. FDD$_\beta$ weights terms based on two relevancy scores. The first score is referred to as *descriptive relevance* (DESCR) and represents the importance of a term to describe the topic. Given a term $t_i$ and a topic $T_k$ the DESCR score is expressed

as:

$$\text{DESCR}(t_i, T_k) = \frac{|d_j : t_i \in d_j \wedge d_j \in T_k|}{|d_j : d_j \in T_k|}.$$

In the above formula $t_i \in d_j$ stands for the term $t_i$ occurring in the document $d_j$, while $d_j \in T_k$ stands for the document $d_j$ being relevant to the topic $T_k$. The second score represents the *discriminative relevance* (DISCR). This score is global to the collection and is computed for a term $t_i$ and a topic $T_k$ as follows:

$$\text{DISCR}(t_i, T_k) = \frac{|d_j : t_i \in d_j \wedge d_j \in T_k|}{|d_j : t_i \in d_j|}.$$

The $\text{FDD}_\beta$ score combines the DESCR and DISCR scores as follows:

$$\text{FDD}_\beta(t_i, T_k) = (1 + \beta^2) \frac{\text{DISCR}(t_i, T_k) \times \text{DESCR}(t_i, T_k)}{(\beta^2 \times \text{DISCR}(t_i, T_k)) + \text{DESCR}(t_i, T_k)}.$$

The tunable parameter $\beta$ is a positive real factor that offers a means to favor descriptive relevance over discriminative relevance (by using a $\beta$ value higher than 1) or the other way around (by using a $\beta$ value smaller than 1). Human-subject studies reported by *Maisonnave et al. (2021a)* indicate that a $\beta = 0.477$ offers a good balance between descriptive and discriminative power, with a Pearson correlation of 0.798 between relevance values assigned by domain experts and those assigned by $\text{FDD}_\beta$. Note that $\text{FDD}_\beta$ is derived from the $F_\beta$ formula, known as F-score or F-measure, traditionally used in information retrieval, where $\beta$ is chosen such that $\beta > 1$ assigns more weight to recall, while $\beta < 1$ favors precision. While we adopt $F_\beta$ as the term-weighting scheme in our framework, other supervised or unsupervised weighting schemes such as those investigated in (*Moreo, Esuli & Sebastiani, 2018*) can be naturally used to guide the selection of terms from topic-relevant sentences.

- **Step 3. Detect ongoing events from the topic-relevant sentences.** Event Detection (ED) is the task of automatically identifying event mentions in text (*Zhang, Ji & Sil, 2019*; *Nguyen & Grishman, 2018*). An event mention is represented by an event trigger, which is the word that most clearly expresses the occurrence of the event. A specific ED task is Ongoing Event Detection (OED), where the goal is to detect ongoing event mentions as opposed to historical, future, hypothetical, or other forms or events that are neither fresh nor current. The rationale behind focusing on ongoing events only is based on the need of building time series of events with the ultimate goal of learning a causal graph. Therefore, it is required that the detected events are ongoing events at the moment they are reported in the news. In previous work, we defined and extensively evaluated the OED task (*Maisonnave et al., 2021b*). Also, we publicly released a dataset consisting of 2,000 news extracts from the NYT corpus containing ongoing event annotations (*Maisonnave et al., 2019*). A model based on a recurrent neural network architecture that uses contextual word and sentence BERT embeddings (*Devlin et al., 2018*) demonstrated to be highly effective in the OED task, achieving an F1-score of 0.704 on the testing set. In that previous work, we used a pre-trained BERT (Bidirectional Encoder Representations from Transformers) model to build the word and sentence embeddings. BERT is a transformer-based deep language model used for NLP. We built

[1]GloVe vectors pretrained on an English corpus (https://spacy.io/models/en#en_core_web_lg)

the BERT word embeddings using the sum of the last four layers of the BERT pre-trained model. Similarly, the BERT sentence embeddings were built by adding the BERT word embedding for all the words in the sentence.

- **Step 4. Construct event phrase embeddings.** An event-phrase embedding representation based on GloVe vectors [1] (with dimension 300) is built for each event trigger $e_k$ in each sentences or phrase $P = w_1, w_2, \ldots, w_n$ as follows:

$$\text{EPER}(e_k, P) = \sum_{w_i \in P} \frac{1}{(|k - i| + 1)^2} \cdot \text{GloVe}(w_i), \tag{1}$$

  where the event trigger $e_k$ is equal to the word $w_k$ for some $k$, $1 \leq k \leq n$. The EPER representation allows to create a phrase embedding that accounts for the GloVe representation of each word $w_i$ in $P$ with a quadratic penalization based on the distance of $w_i$ to $e_k$.

- **Step 5. Group events into semantically cohesive clusters.** Clustering is applied to group similar event-phrase embeddings. Clustering event-phrase embeddings rather than clustering event triggers makes it possible to group events that have a similar representation. This overcomes the problem of dealing with lexically different event mentions that are conceptually associated as independent entities with no relation to each other. Since the number of event representations is typically very large, the highly efficient *MiniBatch KMeans* (*Sculley, 2010*) algorithm is applied for clustering. However, other efficient variants of the *KMeans* algorithm, such as the one presented in *Kanungo et al. (2002)* could be applied. A heuristic such as the *Elbow* method (*Thorndike, 1953*) is applied to determine the number of clusters.

- **Step 6. Build a dataset of relevant variables observed over time.** A dataset is constructed containing measurements of the observations of terms and event clusters occurrences across time.

- **Step 7. Construct a time series of relevant variables.** A time series is generated with the terms and event clusters at the desired temporal granularity (*e.g.*, monthly, weekly, daily, *etc.*).

- **Step 8. Learn a causal structure for the given variables.** A causal graph is learned where the nodes are the variables (terms and event clusters) identified in steps 2 and 5. The edges of the graph are the causal relations learned by applying a causal structure learning technique to the time series generated in Step 7.

It is worth mentioning that the proposed framework has several parameters that potential users could adjust to tailor it to the specific user needs. In step 1, the method adopted to filter sentences relevant to a topic is up to the user (*e.g.*, querying a search engine, string-match filtering, *etc.*). In Step 2, the user can configure the $\beta$ value according to the specific needs. In Step 5, the user should analyze different $K$ values for the KMeans algorithm to choose the one that better suits the use case under analysis. Steps 1 through 5 provide the user with candidate variables to include in the causal graph. The user can manually inspect them and include all of them or only a subset. Lastly, in Step 7, the user might want to choose the level of granularity for the time series (*i.e.,* monthly, weekly, daily, *etc.*).

## APPLYING THE FRAMEWORK

This section describes the application of the proposed framework to a case study and evaluates its performance through a user study with domain experts.

### Case study

The full NYT corpus, covering a period of 246 months (roughly 20 years), is used as a source of news articles. The topic "Iraq" is chosen to illustrate the application of the framework. Iraq was selected for the case study because it represents the geopolitical entity (GPE) outside the United States with the highest number of mentions in the analyzed corpus based on the spaCy's named entity recognizer (https://spacy.io/). The rationale for choosing a GPE outside the United States is that it allows to carry out a more focused and coherent analysis. Note, however, that any other topic, including another GPE, organization name, person name, or economic, social, political, or natural phenomenon with a sufficiently large number of mentions in the corpus could be chosen as a case study. The application of the framework is presented next.

- **Step 1.** For the sake of simplicity we assume that a topic is characterized by an n-gram or set of n-grams. A sentence is said to be relevant to a topic if it contains a mention of any of the n-grams associated with the topic. Since in this case study, we use the GPE "Iraq" as the description of the topic of interest, all the sentences containing the term "Iraq" are selected from the NYT corpus, resulting in 180,206 mentions in 170,497 unique sentences.

- **Step 2.** The $FDD_\beta$ score was used to weight, rank, and select terms from the set of topic-relevant sentences. Note that since $FDD_\beta$ is a supervised term-weighting technique, a sample of sentences non-relevant to the given topic is also needed. Consequently, 170,497 non-relevant sentences (the same number as relevant sentences) were randomly collected from the NYT corpus. Finally, ten topic-relevant terms are selected by applying the $FDD_\beta$ scheme with $\beta = 0.477$ to the set of relevant and non-relevant sentences, resulting in the list of terms presented in Table 1.

- **Step 3.** A total of 498,560 ongoing event mentions are detected by applying the OED task on the 170,497 sentences related to "Iraq" selected in Step 1.

- **Step 4.** An event-phrase embedding representation is built for each event mention detected in Step 3.

- **Step 5.** *MiniBatch KMeans* is applied to group the 498,560 event-phrase embedding representations built in Step 4 into 1,000 clusters. The value $K = 1,000$ is selected by applying the *Elbow* method (*Thorndike, 1953*). Only six highly cohesive clusters with a clear semantic and containing a large number of event mentions are selected to define event cluster variables. The event clusters selected for this analysis are described in Table 2.

- **Step 6.** Measurements of the observations of the sixteen selected variables (ten terms and six event clusters) across time are collected in a dataset.

- **Step 7.** A monthly time series of length 246 (January 1987–June 2007) of the sixteen selected variables (ten terms and six event clusters) is built. Note that a different

**Table 1  Terms identified during Step 2.**

| Term |
| --- |
| "weapons mass destruction" |
| "Persian Gulf war" |
| "United Nations Security" |
| "Iraq invasion Kuwait" |
| "chemical biological weapons" |
| "military action Iraq" |
| "United States" |
| "war Iraq" |
| "Saddam Hussein" |
| "Bush administration" |

granularity (*e.g.*, weekly or daily) or a different number of variables could be used to build the time series. As an example, we present in Fig. 3A the resulting time series built by the framework for the events Military Action (C249) and Death Reports (C109). Another example of time series for the terms "military action iraq", "Iraq invasion Kuwait", and "chemical weapons" is presented in Fig. 3B. To get additional insight into the characteristics of the generated time series, we checked the stationarity of the variables. In the first place, we performed Augmented Dickey–Fuller (ADF) unit root tests for stationarity on each time series. From the ADF tests, we conclude that all the analyzed series are stationary, except for the one corresponding to the variable "bush administration". To look further into this non-stationary time series we applied the Zivot-Andrews (ZA) unit root test and concluded that the series is stationary with a structural break. Note that the variable "bush administration" is ambiguous since it refers to both the administrations of the 41st and the 43rd presidents of the US. But we can show that the structural break clearly distinguishes them. On the other hand, the number of observations corresponding to the first Bush presidency is almost null. That difference in the time series behavior before and after 2001 is likely the reason why the series is not stationary unless you count in the structural change that happened around that time. According to the ZA test, the structural change happened in December 2000. The results of the ADF and ZA tests are presented in Fig. 4.

- **Step 8.** Causal relations are learned from the time series by applying any of the causal structure learning technique described earlier.

The causal graph resulting from applying the described framework to the NYT corpus on the topic "Iraq" is presented in Fig. 5. The *ensemble*$_\cap$ technique was used in this example because high precision is desired to only include causal relations with high confidence. Note, however, that any of the previously described causal learning techniques or other ensemble approaches could be adopted (*e.g.*, based on a weighted voting scheme).

By analyzing the resulting causal graph it is possible to identify several causal relations with a clear semantic. For instance, the causal link "weapons mass destruction" → "military action Iraq" represents the possible existence of *weapons of mass destruction*

**Table 2** Event clusters identified during Step 5.

| Cluster | Salient terms | Description |
|---------|---------------|-------------|
| C109 | killed, Iraq, American, soldiers, civilians | Death reports |
| C165 | against, war, Iraq, opposed, threat | Negative connotation reports |
| C201 | attacks, terrorist, Iraq, missile, suicide | Terrorist attack reports |
| C249 | attack, Iraq, military, missile, against | Military actions |
| C269 | invasion, Iraq, Kuwait, American | Kuwait invasion |
| C550 | war, Iraq, led, 2003 | Iraq war |

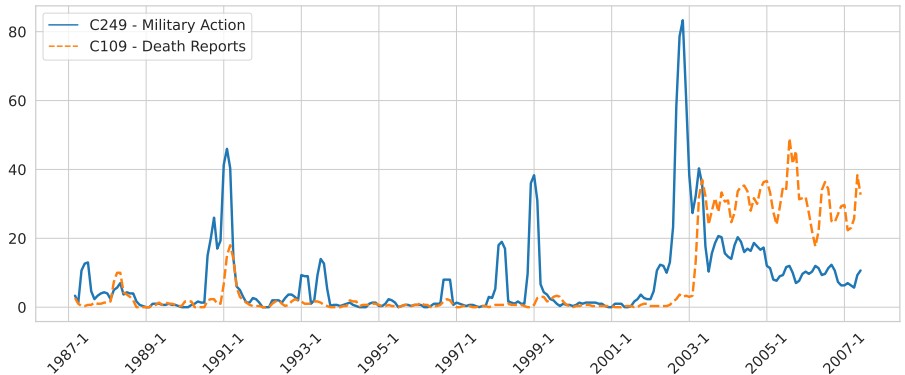

(a)   Ongoing event time series

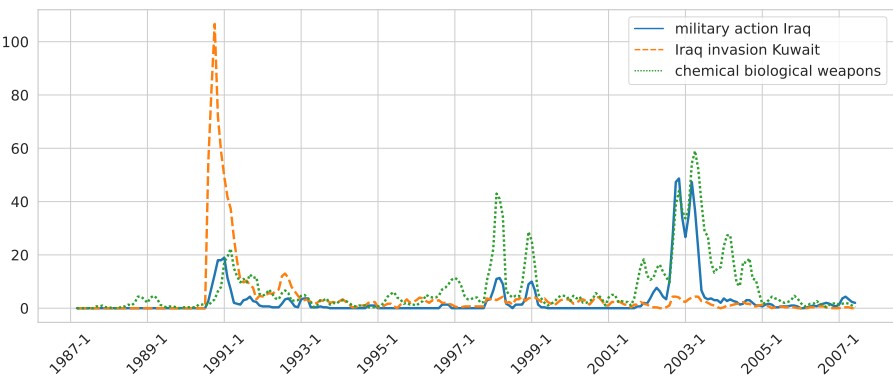

(b)   Term time series

**Figure 3** **Example of ongoing event time series (A) and term time series (B) associated with the topic "Iraq" extracted from the NYT corpus by the proposed framework.**

*in Iraq* as a possible reason for *initiating military actions*. Another causal link between *military actions* and the *war in Iraq* is represented by "`military action Iraq`" → "`war Iraq`". Although these relations should not be automatically interpreted as an actual causal relation, they offer valuable information on pairs of variables with strong co-occurrence where one precedes the other. Also, the causal link "`war Iraq`" → C109, where the variable

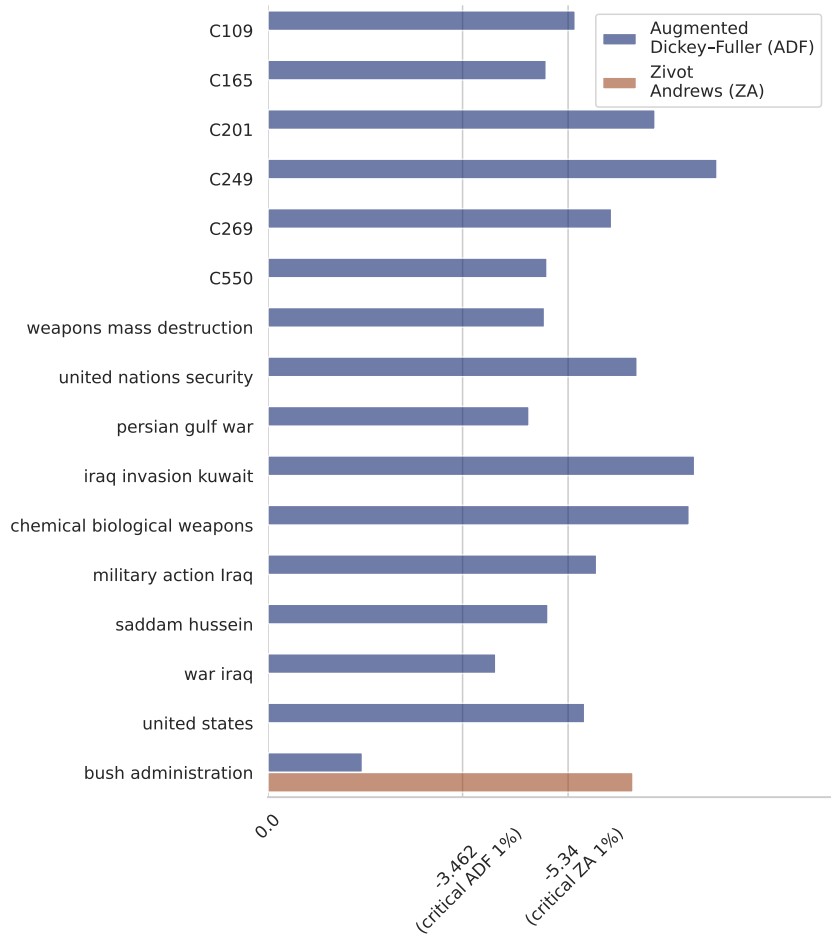

**Figure 4   Unit root tests.** According to the critical values for the ADF test all the series are stationary, except for "Bush administration". The ZA unit root test indicates that the series is stationary with a structural break in December 2000.

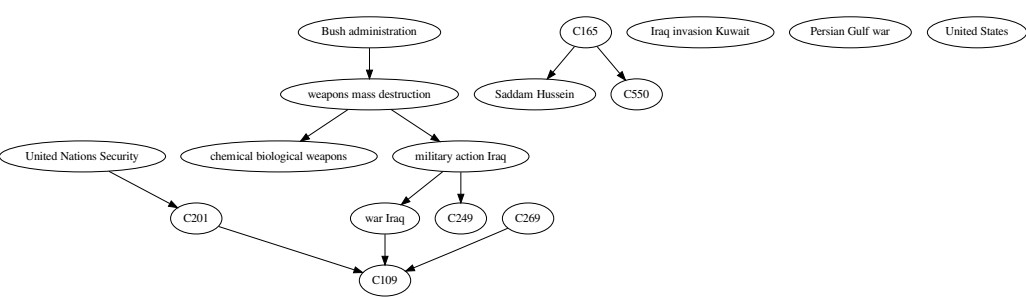

**Figure 5   Causal graph resulting from applying the *ensemble*∩ technique over the time series built from the NYT corpus on the topic "Iraq".**

C109 represents *death reports*, offers an intuitively correct causal relation. Other highly intuitive causal links identified by the framework are "weapons mass destruction" → "chemical biological weapons" and C201 → C109 ← C269, with C201 and C269 representing *terrorist attack reports* and *mentions to the Kuwait invasion*, respectively.

While not all the causal links identified by the framework are necessarily correct, they provide useful information on potential causal relations in a domain. The next section presents an evaluation by domain experts of different casual relations inferred by the framework by applying the most promising causal structure learning techniques.

## Evaluation

The case study presented in the previous section offers initial evidence on the utility of the proposed framework. However, a systematic evaluation is required to provide stronger evidence of the effectiveness of the proposed approach. The evaluation methodology adopted in this work consists of two major evaluation tasks outlined in Fig. 6: (1) an evaluation with synthetic data from two well-known datasets (*TETRAD* and *CauseMe*) and (2) an evaluation with real-world data generated by the proposed framework (based on the case study on the topic "Iraq" described earlier). The following two sections describe each of the evaluation tasks. The first evaluation task addresses the first research question (*i.e.,* **RQ1.** *What methods for time-series causality learning are effective in generalized synthetic data?*). The second evaluation task offers evidence to answer the second research question (*i.e.,* **RQ2.** *Which of the most promising methods for time-series causality learning identified through RQ1 are also effective on real-world data extracted from news?*). Finally, the second evaluation task also addresses the third research question (*i.e.,* **RQ3.** *What type of variables extracted from a large corpus of news is effective for building interpretable causal graphs on a topic under analysis?*).

### Evaluation on synthetic data

For the experiments carried out with synthetic data, two different sources are used: (1) *TETRAD* (*Scheines et al., 1998*) and (2) *CauseMe* (*Runge et al., 2019a*). The simulation tool *TETRAD* was used to generate 56 synthetic datasets with different characteristics. In addition, the eight datasets corresponding to the eight experiments of the nonlinear-VAR datasets (https://causeme.uv.es/model/nonlinear-VAR/) were selected from the *CauseMe* benchmarking platform (causeme.net), resulting in a total of 64 synthetic datasets.

The *TETRAD* datasets were generated by varying the configuration parameters such as the time series length (T ∈ {100, 500, 1000, 2000, 3000, 4000, 5000}), number of observed variables (N ∈ {6, 9, 12, 15, 18, 21, 24, 27, 30}), number of hidden variables (H ∈ {0, 2, 4, 6, 8, 10, 12}), and number of lags (L ∈ {1, 2, 3, 4, 5}). In addition, two settings were used to build the DAG, namely *scale-free DAG* (*SFDAG*) and *random forward DAG* (*RFDAG*). The average performance (across all the evaluated configuration parameters) of the analyzed state-of-the-art, ensemble, and baseline techniques in terms of precision, recall, and F1-score is presented in Fig. 7.

The results show that the five state-of-the-art techniques that achieved the best precision both for *RFDAG* and *SFDAG* are (from best to worst) *BigVAR*, *Direct-LiNGAM*, *PCMCI*,

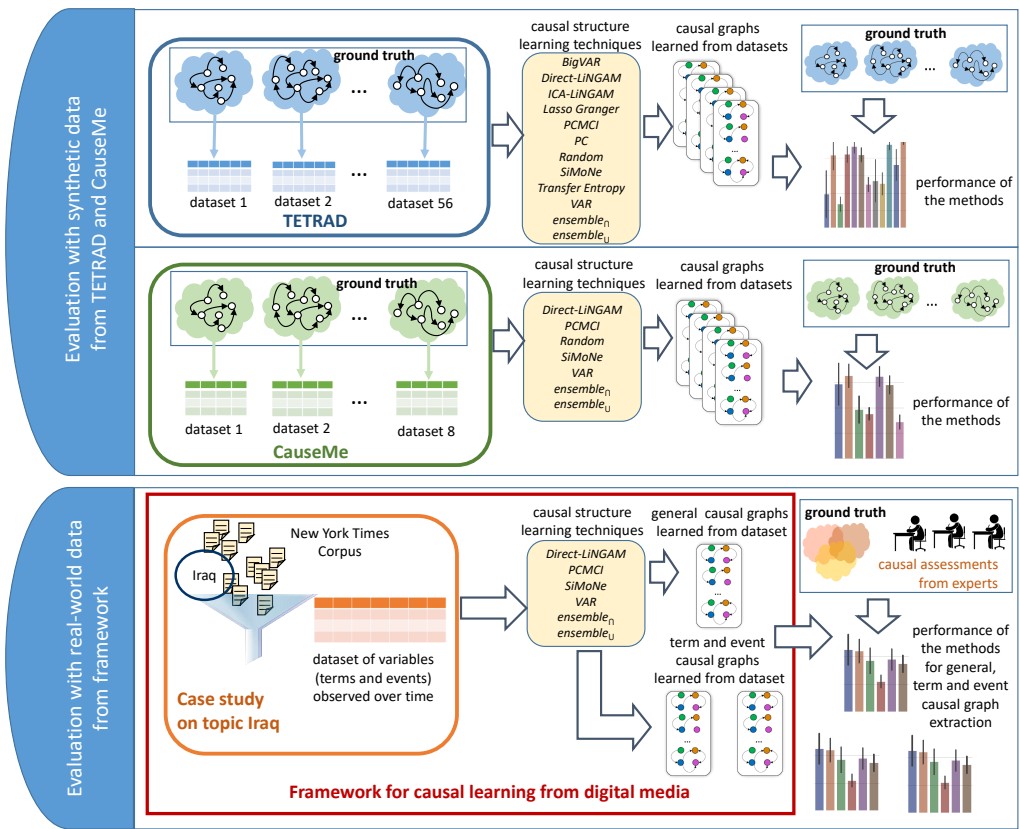

**Figure 6** **Evaluation methodology applied to answer the posed research questions.** The evaluations are conducted on synthetic data (from *TETRAD* and *CauseMe*) and real-world data (generated by the framework and assessed by domain experts). The experiments with synthetic data attempt to identify the most promising time-series causality learning techniques (**RQ1**). The experiments with real-world data look into the question of which of the most promising methods for time-series causality learning identified through the experiments with synthetic data are also effective on real-world variables extracted from news (**RQ2**). The real-world data experiments also investigate what type of variables (terms or events) extracted from news are the most effective ones for building interpretable causal graphs (**RQ3**).

*VAR*, and *PC*. The evaluation in terms of recall shows that the best four state-of-the-art techniques for *RFDAG* are *VAR*, *PCMCI*, *Direct-LiNGAM*, and *PC*. In the case of *SFDAG*, the four best state-of-the-art techniques are the same, but in a slightly different order, namely *VAR*, *PCMCI*, *PC*, and *Direct-LiNGAM*. Finally, the four state-of-the-art techniques that achieved the best F1-score for both *RFDAG* and *SFDAG* are *Direct-LiNGAM*, *PCMCI*, *VAR*, and *PC*. As mentioned earlier, the best four state-of-the-art techniques are combined into two ensemble techniques called *ensemble*$_\cap$ and *ensemble*$_\cup$. Note that *ensemble*$_\cap$ adds a causal relation only when all the combined techniques agree on including it and therefore it tends to favor precision. On the other hand, *ensemble*$_\cup$ adds a causal relation when any of the combined techniques includes it, and as a consequence, it tends to favor recall. Note that *ensemble*$_\cap$ is the technique achieving the best F1-score for both *RFDAG* and *SFDAG*.

The four state-of-the-art techniques that consistently achieved the best performance on *TETRAD* (*Direct-LiNGAM*, *PCMCI*, *VAR*, and *PC*) are further analyzed on the *CauseMe*

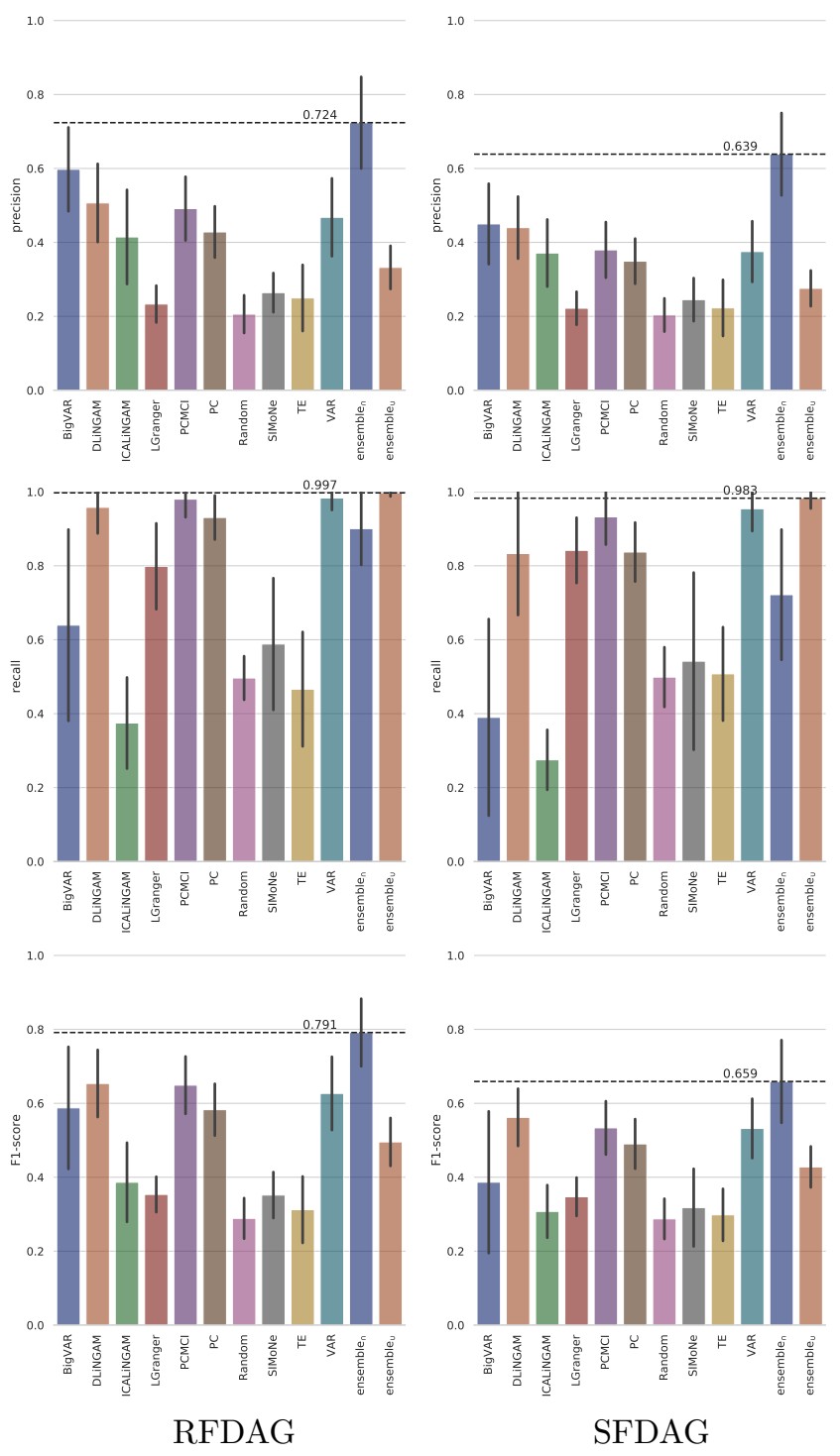

**Figure 7** Averaged performance in terms of precision, recall and F1-score on the *TETRAD* datasets for the evaluated state-of-the-art, ensemble, and baseline techniques. Results are reported both for *RFDAG* (left) and *SFDAG* (right). Confidence intervals are reported at the 95% level.

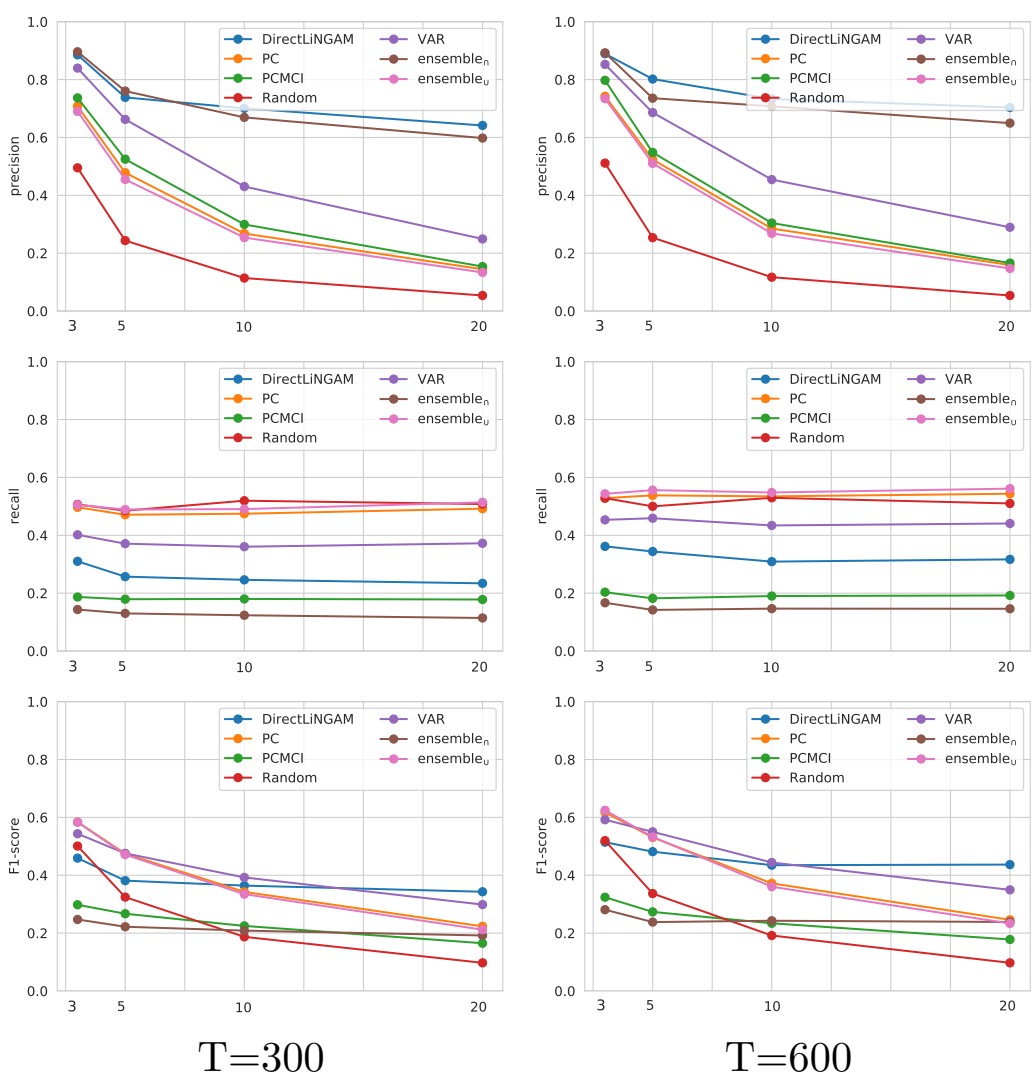

T=300                T=600

**Figure 8** **Performance in terms of precision, recall and F1-score on the *CauseMe* datasets for the evaluated state-of-the-art, ensemble, and baseline techniques.** Results are reported both for time series of length 300 (left) and 600 (right), and for graphs with three, five, 10, and 20 nodes.

datasets. Although *BigVAR* achieves high precision, its confidence intervals for the other metrics are very large, pointing out to inconsistent performance, and therefore it was omitted from the rest of the analysis. The performance achieved by the *Random* technique on the *CauseMe* datasets is also reported for comparison purposes. The eight datasets selected from the *CauseMe* benchmarking platform are built in a similar way with different time series lengths ($T \in \{300, 600\}$) and number of nodes ($N \in \{3, 5, 10, 20\}$). The precision, recall, and F1-score values achieved by the evaluated state-of-the-art, ensemble, and baseline techniques on the eight datasets are reported in Fig. 8. The charts on the left- and right-hand sides present the results for $T = 300$ and $T = 600$, respectively. Each chart displays the results for the four analyzed values of N.

The analysis on *CauseMe* shows a decrease in precision as the number of nodes increases, with *Direct-LiNGAM* and *ensemble*$_\cap$ being the techniques less affected by this loss of performance. On the other hand, the number of nodes does not have a noticeable impact on recall. It is worth mentioning that the high recall values achieved by *Random* are due to the fact that the ground truth causal graph is sparse and *Random* adds edges with a probability of 0.5. It is possible to observe that as the number of nodes increases, the analysis based on F1-score ranks the state-of-the-art techniques (from best to worst) as follows: *Direct-LiNGAM*, *VAR*, *PC*, *PCMCI*, and *Random*.

The evaluation carried out on the *TETRAD* and *CauseMe* datasets provide evidence to address **RQ1** pointing out to the effectiveness of *Direct-LiNGAM*, *VAR*, *PC*, and *PCMCI* for time-series causality learning in generalized synthetic data. We also observe that *ensemble*$_\cap$ tends to achieve high precision while *ensemble*$_\cup$ tends to achieve high recall.

## Evaluation with real-world data

An evaluation is carried out using real-world data generated by the framework based on the case study on the topic "Iraq" described earlier. Three volunteer domain experts (annotators from now on) were recruited for an experiment aimed at assessing the existence of causal relations between pairs of variables extracted by the framework. Two of the annotators had a Ph.D. in History while the third had a Ph.D. in Political Science. Let $T$ and $E$ be the set of terms and event clusters from Tables 1 and 2, respectively. Three sets of unordered pairs of variables of different types (terms and event clusters) were built as follows:

- $P_{\{E,E\}} = \{\{e_1, e_2\} : e_1 \in E \wedge e_2 \in E \wedge e_1 \neq e_2\}$.
- $P_{\{E,T\}} = \{\{e, t\} : e \in E \wedge t \in T\}$.
- $P_{\{T,T\}} = \{\{t_1, t_2\} : t_1 \in T \wedge t_2 \in T \wedge t_1 \neq t_2\}$.

We randomly selected 15 pairs from each of the sets $P_{\{E,E\}}$ (event-event), $P_{\{E,T\}}$ (event-term) and $P_{\{T,T\}}$ (term-term), resulting in a total of 45 pairs. Based on the concept of causality understood by each annotator and having an understanding of the meaning of the variables by reading the annotation guidelines, the annotators were requested to select (to the best of their understanding) one of the following options for each pair of variables $v_1$ and $v_2$:

1. The variables $v_1$ and $v_2$ are causally unrelated (*i.e.*, $v_1 \not\rightarrow v_2$ and $v_2 \not\rightarrow v_1$).
2. The variables $v_1$ and $v_2$ are causally related in both directions (*i.e.*, $v_1 \rightarrow v_2$ and $v_2 \rightarrow v_1$).
3. The variables $v_1$ and $v_2$ are causally related in one direction (*i.e.*, $v_1 \rightarrow v_2$ but $v_2 \not\rightarrow v_1$).
4. The variables $v_1$ and $v_2$ are causally related in the other direction (*i.e.*, $v_2 \rightarrow v_1$ but $v_1 \not\rightarrow v_2$).

Note that for each of the 45 evaluated pairs $\{v_1, v_2\}$ it was possible to derive two Boolean assessments: (1) $v_1$ causes $v_2$ or $v_1$ does not cause $v_2$ and (2) $v_2$ causes $v_1$ or $v_2$ does not cause $v_1$. As a result, we obtained a total of 90 Boolean labels from each annotator. After collecting the list of labels from each annotator, we measured the inter-annotator agreement by computing the Cohen's Kappa coefficient between each pair of annotators. Since we were interested in investigating whether different types of variables (events *vs.* terms) had a different effect on the analysis, we computed separate coefficients for event-event,

**Table 3** Cohen's Kappa coefficients among annotators.

| Annotator | | All | Eevent-event | Event-term | Term-term |
|---|---|---|---|---|---|
| #1 | #2 | 0.2708 | 0.1250 | 0.1667 | 0.0000 |
| #1 | #3 | 0.2194 | 0.7945 | 0.1026 | 0.0000 |
| #2 | #3 | −0.0657 | 0.0426 | 0.0870 | −0.0714 |
| Average | | 0.1415 | 0.3207 | 0.1188 | −0.0238 |

event-terms and term-term pairs. The resulting Cohen's Kappa coefficients of each type of pairs and the average value are reported in Table 3. We observe that there tends to be no agreement between term-term pairs, while the agreement between event-event pairs is considerably superior to the agreement observed between the other types of pairs. This points to the important fact that ongoing events offer a better ground for causality analysis than terms do. However, we observe little agreement among annotators in general, which indicates that the identification of causal relations is a subjective and difficult problem even for domain experts.

Due to the lack of a reliable gold-standard ground truth derived from domain experts to carry out a conclusive evaluation of the analyzed causal structure learning techniques we built three ground truth approximations as follows:

- **Bold Ground Truth**: variable $v_1$ causes $v_2$ if and only if at least one annotator indicates the existence of a causal relation from $v_1$ to $v_2$.
- **Moderate Ground Truth**: variable $v_1$ causes $v_2$ if and only if the majority of the annotators (*i.e.*, at least two annotators) agree on the existence of a causal relation from $v_1$ to $v_2$.
- **Conservative Ground Truth**: variable $v_1$ causes $v_2$ if and only if the three annotators agree on the existence of a causal relation from $v_1$ to $v_2$.

The effectiveness of each of the analyzed causal discovery methods based on **Bold Ground Truth**, **Moderate Ground Truth**, and **Conservative Ground Truth** is reported in Tables 4, 5 and 6, respectively. As expected, the precision tends to increase as the ground truth becomes less conservative, while the recall is higher for a more conservative ground truth. Also, in the same way as in the evaluations carried out with synthetic data, the highest precision is usually achieved by *ensemble*$_\cap$ (except for **Bold Ground Truth**), while the highest recall is always achieved by *ensemble*$_\cup$. This analysis provides evidence to answer **RQ2**, indicating that *ensemble*$_\cap$ and *ensemble*$_\cup$ are effective for learning causal relations, depending on whether the goal is to achieve high precision or high recall, respectively.

Since the inter-annotator agreement for the assessed causal relations reported in Table 3 indicates that event-event relations offer a better ground for causality analysis, we looked into the performance of each of the analyzed causal structure learning techniques restricted only to pairs from $P_{\{E,E\}}$ (*i.e.*, both variables represent event clusters). The results of this analysis based on **Bold Ground Truth**, **Moderate Ground Truth**, and **Conservative Ground Truth** are reported in Tables 7, 8 and 9, respectively. It is interesting to note that by restricting the analysis to variables representing events only, the performance achieved

**Table 4 Methods' effectiveness on bold ground truth.**

| Method | Accuracy | Precision | Recall | F1-score |
|---|---|---|---|---|
| Direct-LiNGAM | 0.3000 | 0.6667 | 0.1194 | 0.2025 |
| PC | 0.4444 | 0.8400 | 0.3134 | 0.4565 |
| PCMCI | 0.4556 | 0.9091 | 0.2985 | 0.4494 |
| VAR | 0.5000 | 0.8235 | 0.4179 | 0.5545 |
| ensemble$_\cap$ | 0.2778 | 0.7500 | 0.0448 | 0.0845 |
| ensemble$_\cup$ | 0.5778 | 0.8372 | 0.5373 | **0.6545** |

**Notes.**
The best results are shown in bold.

**Table 5 Methods' effectiveness on moderate ground truth.**

| Method | Accuracy | Precision | Recall | F1-score |
|---|---|---|---|---|
| Direct-LiNGAM | 0.6889 | 0.3333 | 0.1667 | 0.2222 |
| PC | 0.7000 | 0.4400 | 0.4583 | **0.4490** |
| PCMCI | 0.7111 | 0.4545 | 0.4167 | 0.4348 |
| VAR | 0.6000 | 0.3235 | 0.4583 | 0.3793 |
| ensemble$_\cap$ | 0.7333 | 0.5000 | 0.0833 | 0.1429 |
| ensemble$_\cup$ | 0.5667 | 0.3256 | 0.5833 | 0.4179 |

**Notes.**
The best results are shown in bold.

**Table 6 Methods' effectiveness on conservative ground truth.**

| Method | Accuracy | Precision | Recall | F1-score |
|---|---|---|---|---|
| Direct-LiNGAM | 0.8000 | 0.1667 | 0.2000 | 0.1818 |
| PC | 0.7000 | 0.1600 | 0.4000 | 0.2286 |
| PCMCI | 0.7333 | 0.1818 | 0.4000 | **0.2500** |
| VAR | 0.6222 | 0.1471 | 0.5000 | 0.2273 |
| ensemble$_\cap$ | 0.8667 | 0.2500 | 0.1000 | 0.1429 |
| ensemble$_\cup$ | 0.5444 | 0.1395 | 0.6000 | 0.2264 |

**Notes.**
The best results are shown in bold.

by most methods tends to be superior when evaluated on **Moderate Ground Truth** and **Conservative Ground Truth**.

The evaluation carried out with real-world data and domain experts points to two important findings. In the first place, inter-annotator agreement on causal relations significantly increases when the variables represent ongoing events rather than general terms. In the second place, the performance of most methods tends to improve when the analysis is restricted to ongoing events. Hence, as an answer to **RQ3** we conclude that variables representing ongoing events extracted from a large corpus of news are more effective for building interpretable causal graphs than variables representing terms.

**Table 7   Methods' effectiveness for event-event causal relations on Bold Ground Truth.**

| Method | Accuracy | Precision | Recall | F1-score |
|---|---|---|---|---|
| Direct-LiNGAM | 0.2333 | 0.6000 | 0.1250 | 0.2069 |
| PC | 0.3667 | 0.7778 | 0.2917 | 0.4242 |
| PCMCI | 0.4667 | 0.9000 | 0.3750 | 0.5294 |
| VAR | 0.4000 | 0.7500 | 0.3750 | 0.5000 |
| ensemble$_\cap$ | 0.2333 | 0.6667 | 0.0833 | 0.1481 |
| ensemble$_\cup$ | 0.5000 | 0.8000 | 0.5000 | **0.6154** |

**Notes.**
The best results are shown in bold.

**Table 8   Methods' effectiveness for event-event causal relations on Moderate Ground Truth.**

| Method | Accuracy | Precision | Recall | F1-score |
|---|---|---|---|---|
| Direct-LiNGAM | 0.5333 | 0.4000 | 0.1538 | 0.2222 |
| PC | 0.6667 | 0.6667 | 0.4615 | 0.5455 |
| PCMCI | 0.6333 | 0.6000 | 0.4615 | 0.5217 |
| VAR | 0.5667 | 0.5000 | 0.4615 | 0.4800 |
| ensemble$_\cap$ | 0.6000 | 0.6667 | 0.1538 | 0.2500 |
| ensemble$_\cup$ | 0.6000 | 0.5333 | 0.6154 | **0.5714** |

**Notes.**
The best results are shown in bold.

**Table 9   Methods' effectiveness for event-event causal relations on conservative ground truth.**

| Method | Accuracy | Precision | Recall | F1-score |
|---|---|---|---|---|
| Direct-LiNGAM | 0.6333 | 0.2000 | 0.1250 | 0.1538 |
| PC | 0.6333 | 0.3333 | 0.3750 | 0.3529 |
| PCMCI | 0.6667 | 0.4000 | 0.5000 | **0.4444** |
| VAR | 0.6000 | 0.3333 | 0.5000 | 0.4000 |
| ensemble$_\cap$ | 0.7000 | 0.3333 | 0.1250 | 0.1818 |
| ensemble$_\cup$ | 0.5667 | 0.3333 | 0.6250 | 0.4348 |

**Notes.**
The best results are shown in bold.

# CONCLUSIONS

This article looked into the problem of extracting a causal graph from a news corpus. An initial evaluation using synthetic data of nine state-of-the-art causal structure learning techniques allowed us to address **RQ1**, offering insight into which are the most promising methods for time-series causality learning. The evaluation with domain experts helped us to respond to **RQ2**, by making it possible to further compare the analyzed methods and to assess the overall performance of the proposed framework using real-world data. The labeling task carried out by experts offered interesting insights into the problem of building a ground truth derived from annotators' assessments. In the first place, we learned from the evaluation that there tends to be little agreement among annotators in general, which points to the high subjectivity in causality analysis. However, we also noticed that

the inter-annotator agreement significantly increases when the variables representing potential causes and effects refer to ongoing events rather than general terms (n-grams). We contend that this results from the fact that assessing causal relations between events is a better-defined problem than assessing causal relations between other variables with unclear semantics, such as general terms. This finding provides an initial answer to **RQ3**.

The lack of ground truth for causal discovery is a limitation recurrently discussed in the literature (*Li, Zhang & Cui, 2019*; *Cheng et al., 2022*) and hence the tendency to use synthetic datasets to evaluate new causal discovery techniques. In this work, we took a further step, building a ground truth dataset for causal analysis in a real-world domain, which albeit its limitations, provides a new instrument for measuring the performance of casual discovery techniques.

The practical implications of this framework can be understood in terms of the analyses that can be derived from causal graphs obtained empirically. In particular, this approach helps identify new or unknown relationships associated with a topic or variable of interest that can offer a new perspective to the problem. Constructing a causal graph could be one of the first steps in building causal models. By moving from purely predictive models to causal modeling, we are enriching the level of analysis that could be performed over the variables and relationships of interest, allowing analysts not only to reason over existing data but to evaluate the effect of possible interventions or counterfactuals that did not occur in the observed data. Such analysis is possible because causal modeling allows us to model the generative process of the data, which leads to more robust and complete models. These practical applications of this framework can be highly relevant for public policy makers and social researchers aiming to evaluate cause and effect relations reported in large text corpora.

While we have evaluated the most salient causal discovery methods from the literature and merged the most effective ones into two ensemble methods, as part of our future work we plan to develop a novel causal discovery method from observational data that combines ideas coming from machine learning and Econometrics. The proposed transformation of data from news into time series of relevant variables makes it possible to combine data coming from news with other variables that are typically available as time series (*e.g.*, stock market data, socioeconomic indicators, among others), enriching the domain and providing experts with additional valuable information. This will be explored as part of our future work.

Also, we plan to integrate the complete framework into a visual tool that will assist users in identifying relevant variables and exploring potential causal relations from digital media. The tool interface will allow the user to adjust different parameters to explore the data in more detail. For instance, the user could decide on the number and type of variables and causal links displayed in the causal graph, the time granularity of the time series (monthly, weekly, daily, *etc.*), and the number of event clusters, among other options. Finally, we plan

to conduct additional user studies to further evaluate whether the developed tool facilitates sense-making in complex scenarios by domain experts.

### Funding
This work was supported by CONICET, Universidad Nacional del Sur (PGI-UNS 24/N051 and PGI-UNS 24/E145), ANPCyT (PICT 2019-01640, PICT 2019-02302, and PICT 2019-03944), a LARA project (Google Research Award for Latin America 2019-2022), a New Frontiers in Research Fund Exploration Grant, an ELAP scholarship by the Department of Foreign Affairs, Trade and Development Canada, Compute Canada, and ACENET. The funders had no role in study design, data collection and analysis, decision to publish, or preparation of the manuscript.

### Grant Disclosures
The following grant information was disclosed by the authors:
CONICET, Universidad Nacional del Sur: PGI-UNS 24/N051, PGI-UNS 24/E145.
ANPCyT: PICT 2019-01640, PICT 2019-02302, PICT 2019-03944.
A LARA project (Google Research Award for Latin America 2019-2022).
A New Frontiers in Research Fund Exploration Grant.
An ELAP scholarship by the Department of Foreign Affairs, Trade and Development Canada, Compute Canada, and ACENET.

### Competing Interests
Ana Maguitman is an Academic Editor for Peer J Computer Science.

### Author Contributions
- Mariano Maisonnave conceived and designed the experiments, performed the experiments, analyzed the data, performed the computation work, prepared figures and/or tables, authored or reviewed drafts of the article, and approved the final draft.
- Fernando Delbianco conceived and designed the experiments, performed the experiments, analyzed the data, performed the computation work, authored or reviewed drafts of the article, and approved the final draft.
- Fernando Tohme conceived and designed the experiments, analyzed the data, authored or reviewed drafts of the article, and approved the final draft.
- Evangelos Milios conceived and designed the experiments, analyzed the data, authored or reviewed drafts of the article, and approved the final draft.
- Ana G Maguitman conceived and designed the experiments, analyzed the data, prepared figures and/or tables, authored or reviewed drafts of the article, and approved the final draft.

### Data Availability
The code used for evaluation on synthetic data is available at Universidad Nacional del Sur: https://cs.uns.edu.ar/~mmaisonnave/resources/causality/code/.

The code used for the supervised term-weighting scheme is available at Google Colaboratory (http://cs.uns.edu.ar/~mmaisonnave/resources/FDD_code).

Code used for the supervised term-weighting scheme is available at Mendeley: Maisonnave, Mariano; Delbianco, Fernando; Tohme, Fernando; Maguitman, Ana (2019), ''Economic Relevant News from The Guardian'', Mendeley Data, V3, doi: 10.17632/yt8j2f3hpp.3

The code used for the ongoing event detection task is available at Google Colaboratory (http://cs.uns.edu.ar/~mmaisonnave/resources/ED_code/)

Data used for the ongoing event detection task is available at Mendeley: Maisonnave, Mariano; Delbianco, Fernando; Tohme, Fernando; Maguitman, Ana; Milios, Evangelos (2020), ''Event Detection Dataset'', Mendeley Data, V1, doi: 10.17632/7d54rvzxkr.1

The annotation guidelines, data, and code used for the evaluation by domain experts is available at Google Colaboratory (https://cs.uns.edu.ar/~mmaisonnave/resources/causality-evaluation).

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
