# Peer review of "Causal graph extraction from news: a comparative study of time-series causality learning techniques"

_PeerJ Computer Science, doi:10.7717/peerj-cs.1066_

## Round 0.1 · original submission · Major Revisions

Based on reviewers’ comments, you may resubmit the revised manuscript for further consideration. Please consider the reviewers’ comments carefully and submit a list of responses to the comments along with the revised manuscript.

Reviewer 1 ·

Basic reporting

A good article which presents a novel framework for extracting causal graphs from digital text media that are selected from a domain under analysis by applying specially developed information retrieval and natural language processing methods. The framework is applied to the New York Times dataset, which covers news for a period of 246 months. The proposed analysis offers valuable insights into the problems of identifying topic-relevant variables from large volumes of news and learning causal graphs from time series.

A. Normally, the Abstract and beginning of an introduction section contain the problems in the existing approaches followed by the solution but in this article, the problem statement is somehow not discussed.

B. In the introduction, the proposed work starts from line 48 then in between (lines 53-55) the works of the other authors came.
- It will be nice to have if existing work is discussed in one place followed by proposed work. In this way, the reader will have a good understanding of the work.

C. The employed dataset is consisting of 246 month period.
- Why not this period is mentioned in years. So that each reader doesn’t have to calculate it.

D. How research questions have been formulated/reached? Normally, this is done after an extensive literature review but in this article, it seems some sort of reverse engineering is being done e.g., research questions are mapped to literature.

E. In the related work section, existing research is being discussed but shortcomings of each work are not highlighted. These shortcomings ultimately lead to research questions.

F. Figure 2 can be improved. At the moment, it is a bit congested. E.g., step 2 and step 5 converge at a point that is not very clear to the reader.

G. Line # 200, a topic is said to be relevant if it contains mention
- This is a big assumption because it is not necessary to have mentioned it in all cases.

H. Line # 221, what is BERT embeddings?

I. Line # 248, the selected topic has the highest number of mentions in the corpus.
- Is this a biased input to the proposed framework? What if the selected topic has the lowest number of mentions? How proposed framework will behave in this scenario?

J. Why scale-free and random forward DAGs are used?

K. The conclusion section contains the summary of the paper in the first two paragraphs which should not be there because this is repeating stuff again and again.

L. In future work, integration of this work with some visual tools is mentioned.
- For example?

Experimental design

Experimental design is aligned with the proposed methodology with enough details to reproduce the results.

For example, the code written in python language with sufficient comments is made public for use:
https://cs.uns.edu.ar/~mmaisonnave/resources/causality/code/
Maisonnave et al 2020 - Event Detection.ipynb - Colaboratory (google.com)
Use case evaluation.ipynb - Colaboratory (google.com)

Validity of the findings

All underlying data have been provided with a well-stated discussion.
Economic Relevant News from The Guardian - Mendeley Data
Maisonnave et al 2020 - FDD paper.ipynb - Colaboratory (google.com)

Reviewer 2 ·

Basic reporting

The paper presents a framework for extracting causal graphs from digital text media. It consist of 8 steps that goes from analyzing the text of news to filter topical-relevant sentences, discover events and construct time series to learn a causal structure among variables. The framework is a valuable contribution for the area as it describe all the process starting from raw text to achieve an actual graph structure of events, providing a concrete technique for each of the steps so that it can be effectively operationalized.

The paper is in general well-written and explained. There are some issues that can be improved regarding organization. First, the place of the section “Causal structure learning” is a little misleading, since such learning is just a part of the framework and the last step (not yet introduced at that point), also only mentions existing state-of-the-art algorithms, I think it can be place in the description of step 8 or later. Second, in each step of the framework, although a technique is chosen and described, it would be interesting to mention other alternatives that can take their place in the framework and justify the selection. Other details:

- in step 1 it is mention the use of n-grams, but the examples only have single terms. (e.g. United, States). Why not 2-grams for instance? or NER entities?
- the role of Beta within the framework in step 2 should be clarified, which is the rationale of its value in the framework? In this setting, it is preferred descriptive or discriminative terms?
- in step 3, the event trigger consist of a single word? Which is the effect of using a unique word in this context of a huge volume of texts?
- step 5, the elbow method provide a suggestion for the number of clusters, it this done automatically? All clusters are considered or some small ones can be discarded?
- in the case study, it said “sixteen variables”, should be 6?

Experimental design

The experimentation with a framework such as the one proposed by the authors is difficult because of the lack of ground truth. Therefore, the evaluation with the Iran case study, which was used previously for illustrating the example is valuable. I think the data generated for the case study should be also considered a contribution of the work.
The evaluation with synthetic data of the causal learning technique is again a little out of the focus of the paper, as it only proves which learning algorithms works well with the generated synthetic data, but it is not really the type of data used in the framework. Therefore, it looks more a general evaluation of algorithms unrelated to the framework itself.

Validity of the findings

The findings are interesting and demonstrate that the framework can be operationalized and used to analyze real data. Some additional comments to help to highlight the paper contribution:

- the practical implications of the framework can be discussed, probably in the conclusions. As it can be a valuable tool for analysis, potential scenarios and uses can be identified.
- another aspect to discuss is the level of user intervention that is needed for using the framework, the iterations required and setting of parameters, also some guidelines for their application can be provided
- as the paper introduced some research questions at the beginning it would be important to go back to them in the discussion/conclusions to summarize the findings

---

## Round 0.2 · Minor Revisions

Congratulations, the reviewers are mostly satisfied with the revised version of the manuscript and one has recommended acceptance. However, some of the minor comments have been provided too by a reviewer. Therefore, before submitting the final version please do address those.

Reviewer 1 ·

Basic reporting

Looking good now.

Experimental design

Ok.

Validity of the findings

Fine.

Additional comments

12 questions were raised related to basic reporting. In the revision, most of them are addressed. I think, document is good to go now.

Reviewer 2 ·

Basic reporting

The paper presents a framework for extracting causal graphs from digital text media, considering the analysis of big volumes of text. The contribution is interesting and the resulting framework provides a good starting point for experimentation with causal learning from social media.

The revised version of the paper has been improved regarding clarity of the contribution, analysis of research questions and implications of the work. I still think that the related works section should highlight further the advantages of the approach and potential applications which are not covered by previous works in the literature.

The description of the main steps of the frame work have also been improved, making more clear the scope of the individual techniques and their alternatives.

Experimental design

The experimental evaluation performed with real world data is valuable and provide evidence that the framework can perform well when applied to real data. The authors also clarify the role of the evaluation of synthetic data as a first step to discard some options.

Validity of the findings

The findings have been validated with the experimental evaluation on real-world data. The revised version also discuss the conclusions regarding the research questions as asked and the practical implications of the work presented.

Additional comments

The authors have addressed all my concerns in this revised version, I think the paper has been considerably improved.

---

## Round 0.3 · accepted · Accept

Congratulations, the revised version of the manuscript is satisfactory and it is recommended for publication.

[#